# Sample preparation and data collection for serial block face scanning electron microscopy of mammalian cell monolayers

**Noelle V. Antao**[1¤], **Joseph Sall**[2], **Christopher Petzold**[2], **Damian C. Ekiert**[1,3¤], **Gira Bhabha**[1¤]*, **Feng-Xia Liang**[1,2]*

1 Department of Cell Biology, New York University School of Medicine, New York, NY, United States of America, 2 Office of Science and Research Microscopy Laboratory, New York University School of Medicine, New York, NY, United States of America, 3 Department of Microbiology, New York University School of Medicine, New York, NY, United States of America

¤ Current address: Department of Biology, Johns Hopkins University, Baltimore, MD, United States of America

* gira.bhabha@gmail.com (GB); fengxia.liang@nyulangone.org (FXL)

**Data Availability Statement:** All relevant data are within the paper and its Supporting Information files. SBF-SEM datasets are available in the EMPIAR database: EMPIAR-11683

## Abstract

Volume electron microscopy encompasses a set of electron microscopy techniques that can be used to examine the ultrastructure of biological tissues and cells in three dimensions. Two block face techniques, focused ion beam scanning electron microscopy (FIB-SEM) and serial block face scanning electron microscopy (SBF-SEM) have often been used to study biological tissue samples. More recently, these techniques have been adapted to *in vitro* tissue culture samples. Here we describe step-by-step protocols for two sample embedding methods for *in vitro* tissue culture cells intended to be studied using SBF-SEM. The first focuses on cell pellet embedding and the second on *en face* embedding. *En face* embedding can be combined with light microscopy, and this CLEM workflow can be used to identify specific biological events by light microscopy, which can then be imaged using SBF-SEM. We systematically outline the steps necessary to fix, stain, embed and image adherent tissue culture cell monolayers by SBF-SEM. In addition to sample preparation, we discuss optimization of parameters for data collection. We highlight the challenges and key steps of sample preparation, and the consideration of imaging variables.

## Introduction

Volume electron microscopy (vEM) stands out as a powerful tool capable of producing three dimensional (3D) reconstructions of biological samples, spanning from tissues to individual cells. Resolutions resulting from vEM techniques exceed those attainable with super-resolution microscopy, which can achieve sub-100 nm lateral resolution [1–3]. [2, 33]Typically, sub-organellar structural features, such as the cristae of mitochondria, can be well resolved using vEM techniques [4]. In vEM, a series of two-dimensional images are collected, which can subsequently be processed, annotated and reconstructed in three dimensions. Two block-face

**Funding:** We gratefully acknowledge the following funding sources: SSP-2018-2737 (Searle Scholars Program, to G.B.), R01AI147131 (National Institute of Allergy and Infectious Diseases, NIAID, to G.B.), Irma T. Hirschl Career Scientist Award (to G.B.). G. B. is a Pew Scholar in the Biomedical Sciences, supported by The Pew Charitable Trusts (PEW-00033055). The NYU Microscopy Core is partially supported by NYU Cancer Center Support Grant NIH/NCI P30CA016087, and Zeiss Gemini 300 SEM with 3View was purchased with support of NIH S10 OD0019974. The funders had no role in study design, data collection and analysis, decision to publish, or preparation of the manuscript

**Competing interests:** The authors have declared that no competing interests exist

scanning electron microscopy (SEM)-based techniques are predominantly used to image biological samples in 3D: focused ion beam scanning electron microscopy (FIB-SEM) [5] and serial block-face scanning electron microscopy (SBF-SEM) [6]. In both methods, 3D volumes are acquired by serial sectioning of samples, using either a focused ion beam (FIB-SEM) or an ultramicrotome (SBF-SEM) to remove a thin layer of the sample automatically, followed by imaging of the newly exposed sample block surface by SEM. Although both techniques have a similar spatial resolution in the $x$-$y$ dimension (3–5 nm), FIB-SEM has greater resolution in the $z$ dimension for achieving isotropic voxels down to 3–5 nm voxel size for biological samples, due to the ability to cut thinner sections using a focused ion beam [7]. In contrast, the resolution in the $z$ direction is limited in SBF-SEM, due to the fact that the thinnest section that can be reliably cut using an ultramicrotome is ~25 nm in thickness [8]. While the $z$ resolution is higher using FIB-SEM, the milling process is slower, which can, in practice, limit the total area and volume that can be imaged in a given time relative to SBF-SEM. The ultramicrotome/ diamond knife used for sectioning in SBF-SEM makes the cutting of sections about 4 times faster, allowing for the acquisition of total volumes of $10^6$ μm$^3$ [9, 10]. In contrast, volumes analyzed by ion beam milling are typically $10^3$ μm$^3$ [11, 12]. The faster cutting enables efficient imaging of samples with larger volumes by SBF-SEM, and allows users to evaluate the relationship between cells over relatively large fields of view, across all imaged planes [4, 13–15]. It should be noted that the use of plasma FIB (pFIB) with other ion species like argon [16] and xenon [17] may provide finer and faster milling when compared to gallium, thereby increasing the milling speed and surface area analyzed by pFIB-SEM.

The SBF-SEM instrument consists of an SEM, an ultramicrotome mounted within the SEM chamber, and the necessary hardware and software to control image acquisition [6]. The first imaging setup resembling modern SBF-SEM was invented in 1981 by SB Leighton, in which a miniature microtome mounted within the SEM chamber was used to cut and image successive sections from a resin embedded squid fin nerve tissue sample [18]. Limitations of this first setup included the need to remove the sample from the chamber in between sections, in order to carbon coat it for optimal conductivity, and the limited capabilities for storage and analysis of digital image files, given existing technology at the time. The technique was revisited and improved by Denk and Horstmann [6], who used a variable pressure system, which allowed imaging and analysis of non-conductive samples, overcoming the need for carbon coating between sections. Since then, SBF-SEM has been used to examine biological structures at different length scales from micrometers (axons) to nanometers (individual chromosomes and mitochondrial cristae) [13–15].

Historically, SBF-SEM has been extensively used to study biological tissue samples such as brain tissue for neuronal circuit reconstruction [19, 20]. More recently this methodology has been adapted to study the ultrastructure of many other biological tissue samples e.g. bone tissue, developing unicellular organisms and cell monolayers [21–30]. The varying characteristics of different biological samples requires sample preparation and image acquisition to be tailored to the biological sample being studied. Currently there are very few detailed lab protocols [25, 31] available for newcomers to the field, making this technique less accessible. Here, we systematically address the specific challenges associated with sample preparation and imaging of *in vitro* cultured cells by SBF-SEM. We outline all the steps necessary to stain, embed and image *in vitro* cultured cells for SBF-SEM analysis. We also outline the steps necessary for performing correlative light and electron microscopy (CLEM) in order to target specific or rare biological events [23, 32–34] prior to SBF-SEM data collection. We share this comprehensive lab protocol with the goal of improving the accessibility of vEM methodologies with those interested in entering the vEM field.

The steps detailed in the protocol are optimized for the analysis of adherent mammalian cell lines using the Gatan OnPoint BSE detector in a Zeiss Gemini300 VP FESEM equipped with a Gatan 3View automatic ultramicrotome with focal charge compensation (FCC) [35] to reduce sample charge accumulation during imaging acquisition (Fig 1). However, the basic methodologies detailed in this protocol can also be adapted to other cell types and vEM methodologies. Indeed similar fixation methodologies (variations of the Deerinck protocol) have been adapted to analyze other mammalian cell lines, using different SBF-SEM systems [22–25, 36].

## Overview of SBF-SEM

The goal for vEM imaging techniques such as SBF-SEM is to obtain volumetric information of biological samples with the best image quality and at a resolution sufficient to answer the biological question of interest. It is important that the data be of good quality to ensure the reliability and robustness of 3D reconstructions and quantification, and to facilitate automation of data analysis, such as image segmentation. In SBF-SEM imaging, image quality is determined by: 1) image contrast (signal-to-noise), which can be visualized as differences in electron densities; 2) image resolution, and 3) cutting stability, which refers to the uniformity with which each slice is being cut. The important steps that require optimization to ensure reliable

**A**

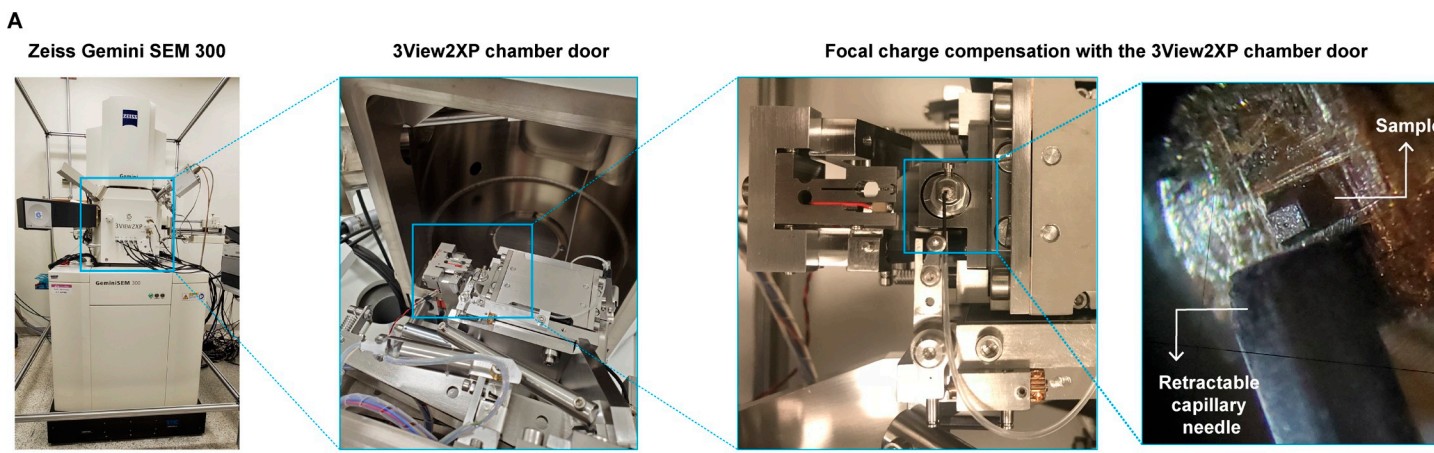

**B**

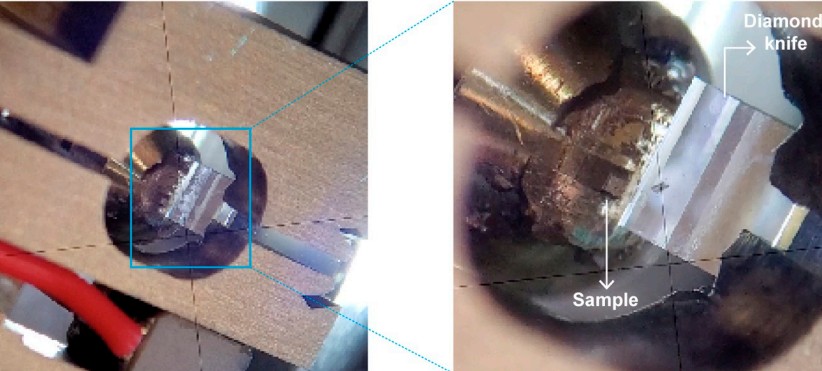

**Fig 1. GeminiSEM 300 with a 3View2XP setup.** (A) 3View2XP stage setup. Insets show the retractable nozzle for the Focal Charge Compensation system (FCC) and the sample block at higher magnification. (B) Picture of the retractable ultramicrotome system in the 3View2XP stage.

generation of volumetric image data by SBF-SEM can broadly be divided into sample preparation and image acquisition.

**Sample preparation.** Imaging biological samples requires special consideration. Biological samples are largely composed of light elements (hydrogen, carbon, nitrogen), making them sensitive to radiation damage and having less contrast under electron beams. Biological materials also contain a significant amount of water, making them incompatible with imaging under high vacuum. Finally, biological samples have low conductivity, making them more susceptible to charging, which distorts images. Therefore, SBF-SEM samples must be prepared such that they preserve the ultrastructure of the biological tissue, have the structural rigidity to withstand the high vacuum and ultramicrotome slicing within the SEM chamber, have sufficient contrast during imaging and optimize sample conductivity to reduce charging artifacts during imaging. The first step in the sample preparation pipeline is tissue fixation and staining to preserve the tissue structure and enhance specimen contrast [37–40]. In this method, the biological sample is treated with chemical fixatives. For example, a mixture of aldehydes such as formaldehyde and glutaraldehyde result in the fixation of proteins via cross-linking [41–43]. Typically, 2–4% formaldehyde and 1–4% glutaraldehyde are used to fix biological samples [22–25]. This is followed by treatment with multiple heavy metal stains simultaneously (known as *en bloc* staining) to enhance the contrast of the sample as well as improve sample conductivity. Osmium tetroxide ($OsO_4$) [44, 45], preferentially interacts with unsaturated lipids, a major component of cell membrane. $OsO_4$ fixes lipids and also acts as an electron stain to improve contrast, thus working as a mordant to enhance lead binding. Thiocarbohydrazide acts as a bridging agent to which more osmium can bind, thereby further enhancing the contrast of lipid components in the cell [38]. Lead aspartate interacts with membranes and proteins [46–48], enhancing contrast of these structures. Uranyl acetate stains nucleic acids and proteins, providing general contrast to biological samples [49, 50]. Given the toxicity and radioactivity of uranyl salts, several replacement compounds have been developed and can be used as alternate contrasting agents. These include two lanthanide salts, samarium triacetate and gadolinium triacetate [51], that have been used to stain plastic embedded samples of plant and animal origin, as well as neodymium acetate [52], used to stain mammalian cells and tissue. Following fixation and staining, samples are dehydrated by incubation in a series of graded organic solvent solutions, such as ethanol, and then embedded in an epoxy resin [53, 54]. The choice of resin impacts how reliably ultra-thin sections can be cut from the block face and should ideally withstand electron beam damage during imaging. There are several options of resins that can be chosen for SBF-SEM sample embedding. These include Durcupan and hard EMbed-812 that perform better in low voltage SBF-SEM imaging when compared to Spurr, LX112 and LR White embedded samples [55]. After the embedding resin has polymerized, the sample blocks are trimmed to a shape amenable for cutting by the ultramicrotome mounted in the SEM chamber. Typically, sample blocks can be trimmed into a pyramid that contains no regions of empty resin i.e. some of the sample is exposed on every side [4, 28, 55]. The sample block face is trimmed to a square or rectangle with a height and width of ~0.5 mm x 0.5 mm. Depending on the biological sample being analyzed, each of the following steps in sample preparation require optimization at the level of: 1) the combination and concentration of heavy metal stains, 2) the incubation time of each stain to ensure even stain penetration throughout the sample, 3) the type of embedding resin and 4) the dimensions of the final sample block that is loaded into the SEM chamber. Specimens should be inspected by TEM for the quality of sample preservation and the presence of any preparation artifacts, such as sample deformation or changes in organelle ultrastructure. Special fixation and sample preparation methods such as high pressure freezing (HPF) and freeze substitution (FS) should be considered to analyze fast cellular events, and for better preservation of cell ultrastructure. HPF-FS

method results in much less sample shrinkage [56–59] and ensures the preservation of samples close to their native state. In this protocol, we discuss the two most popular methods for preparing *in vitro* cultured cell samples using a chemical fixation procedure, cell pellets (Fig 2A and S1 File) and *en face* monolayer cells (Fig 2B and S1 File).

**Image acquisition.** Imaging begins when an electron beam is focused at a point and maintained at that position for an amount of time known as the pixel dwell time. Images are generated by the detection of high energy backscattered electrons (BSE's) emitted from the sample. After the pixel dwell time has elapsed, the beam is moved by one pixel length and the process is repeated in a raster pattern across the block face. Once the entire region of interest is imaged, the sample block is raised by a specified distance, corresponding to slice thickness, in the z-direction. The ultramicrotome then removes a thin section from the surface and the newly exposed blockface is imaged. This sequence of imaging steps is repeated to generate a series of images that are aligned in *x-y* and registered in the z-direction (Fig 3A). Image resolution is influenced by the diameter of the electron beam probe when it hits the sample surface. The primary imaging conditions that control probe diameter are the electron source, accelerating voltage, filament extractor current, and aperture diameter. Increasing the accelerating voltage reduces the probe diameter, which improves spatial resolution in *x-y*. A higher acceleration voltage also increases the penetration and interaction volume of the electron beam within the sample. In the case of non-conductive biological samples, higher accelerating voltages increase the likelihood of sample charging, due to the accumulation of low energy secondary electrons within the sample blockface. Sample charging physically damages the sample blockface, introduces image scanning distortions and reduces image contrast [60]. Several imaging parameters influence image contrast and cut stability during acquisition, including accelerating voltage, aperture size, dwell time, and focal charge compensation [61] (Fig 3B). Varying each of these parameters affects different aspects of the resulting images (Fig 3C). To reduce charging artifacts and damage to the sample, electron dose (beam energy per unit area of the sample) should be kept as low as possible while also considering the signal-to-noise ratio of the scanned image. This can be achieved by lowering the accelerating voltage such that the number of incident electrons is similar to the yield of BSEs and SEs emitted from the sample [62]. For most non-sputter-coated biological samples, charge balance can be achieved around ~1.5 KeV. Charging artifacts can also be mitigated by adjusting the SEM scan settings (line versus frame averaging) and pixel dwell time during image acquisition (Fig 3C). Focus charge compensation (FCC) uses a capillary installed close to the sample block face to direct nitrogen gas onto the sample surface. The ionized nitrogen gas molecules neutralize the accumulated charge on the sample surface[35, 63, 64] (Fig 3C). The amount of nitrogen gas and the chosen pixel dwell time must both be optimized to minimize sample charging without sacrificing image contrast.

To explore the effects of different imaging parameters on image quality, we imaged a sample block of mammalian cells prepared using the cell pellet method, and varying different imaging parameters (Fig 4). Condition 1 was imaged at 2 nm pixel size, using an accelerating voltage of 1.2 keV (Fig 4A). With these imaging conditions, image contrast was good (Fig 4B) and the ultrastructures of host mitochondria and ER well resolved (Fig 4B insets). However, these conditions also caused increased charging in the sample (visible as darker regions in the resin) and cutting artifacts (Fig 4C, cyan arrowheads, Movie 1). To reduce the charging and cutting artifacts observed in condition 1, we first tested the effect of increasing the N2 gas pressure (Fig 4A, condition 2). We observed that this suppressed the charging artifacts, but some cutting artifacts were still visible (Fig 4D, insets, Movie 2). We also tested the effect of reducing the beam dwell time (Fig 4A, condition 3). This suppressed charging and gave the most stable cutting through the 3D volume, with a slight reduction in image contrast (Fig 4E, Movie 3).

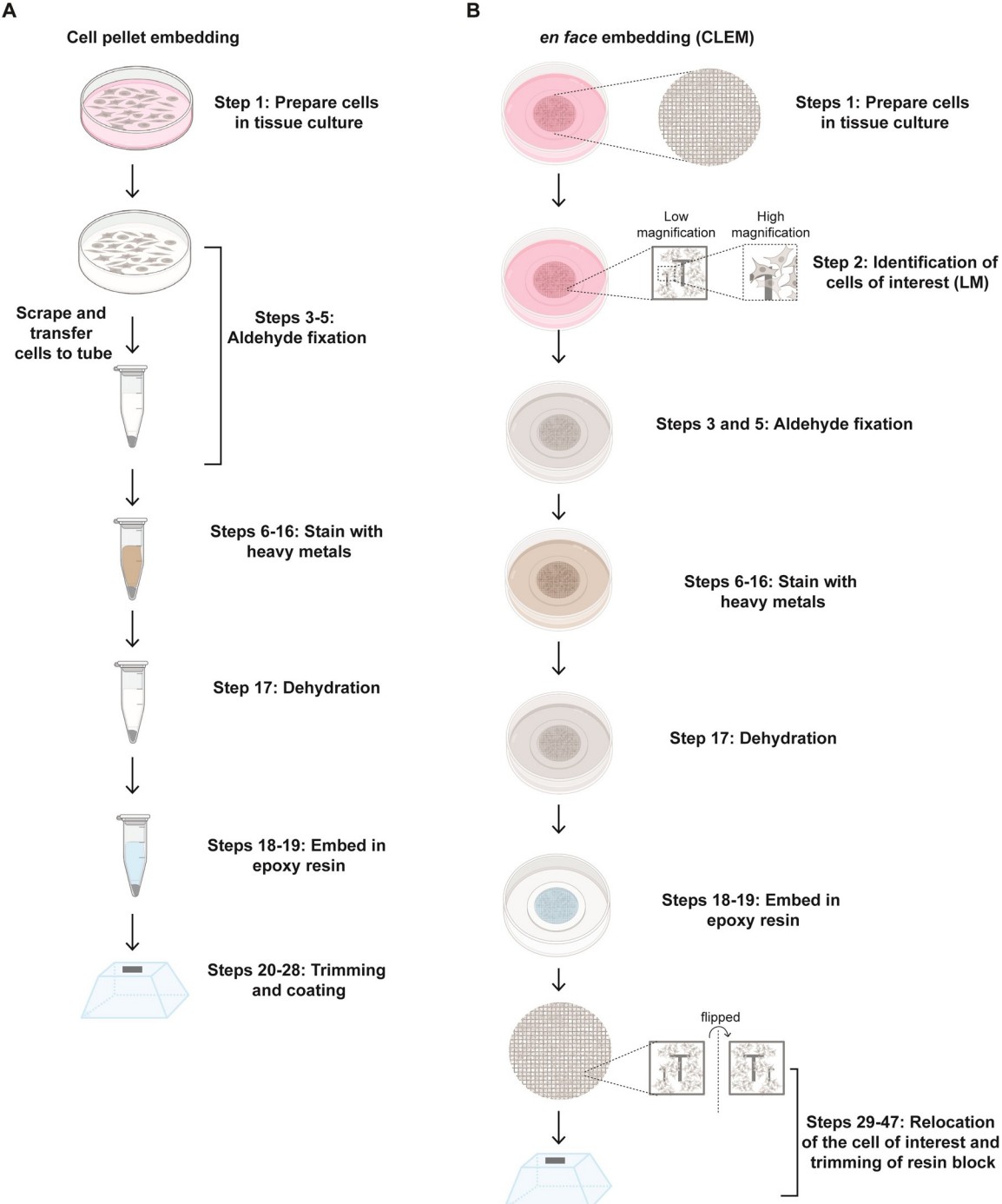

**Fig 2. Key steps in the workflow for SBF-SEM of adherent tissue culture cells.** (A) Cell pellet workflow and (B) *en face* with CLEM workflow. Step numbers on the Fig correspond to step numbers in the step-by-step protocol in the S1 File and on protocols.io (doi.org/10.17504/protocols.io.e6nvwdz5zlmk/v2).

Finally, we tested the effect of increasing the accelerating voltage while also decreasing the aperture size and dwell time as well as increasing the N2 gas pressure in the chamber (Fig 4A, condition 4). With these imaging conditions, we also observed good image contrast and

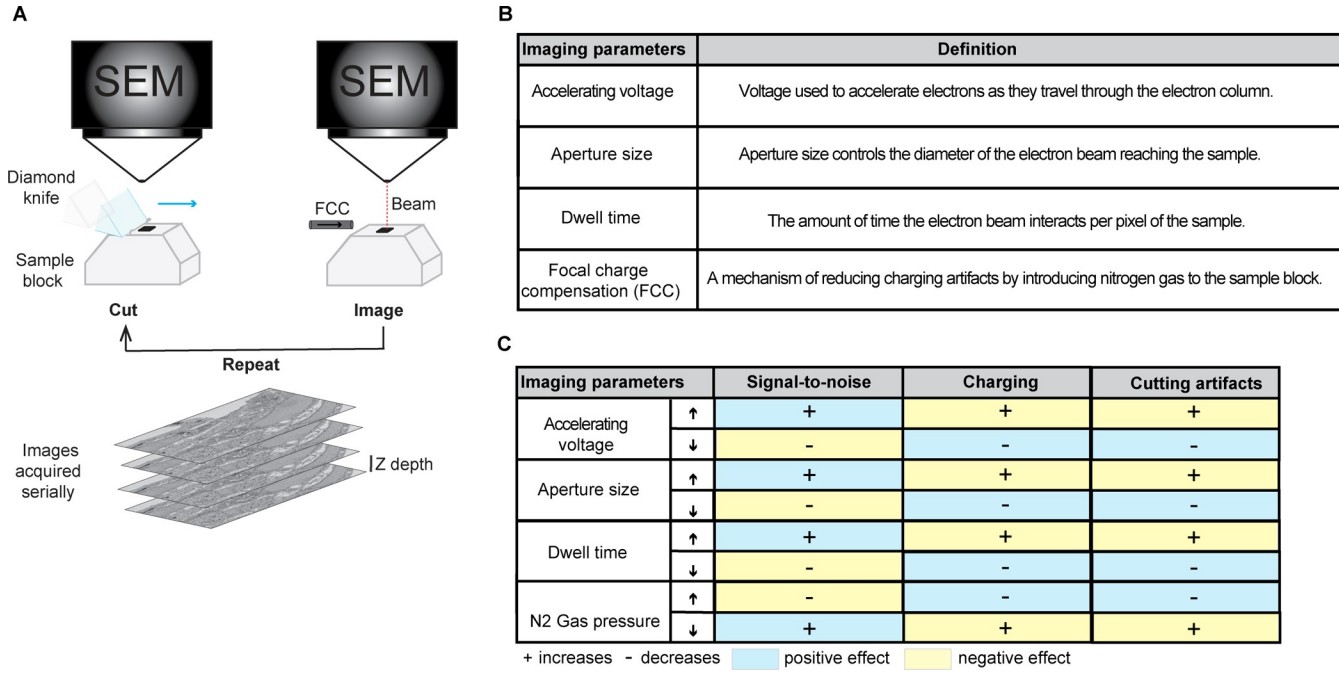

**Fig 3. SBF-SEM imaging concept and key imaging parameters.** (A) Schematic of the SBF-SEM sample and imaging setup. FCC (Focal Charge Compensation). (B) Table of the key imaging parameters and their definitions. (C) Table describing how increasing (up arrow) or decreasing (down arrow) key parameters impacts properties of the resulting image, including signal-to-noise, charging and cutting artifacts during imaging.

preserved cutting stability of the blockface through the sample volume tested (Fig 4F, Movie 4). Therefore, it is important to tailor imaging parameters to identify image acquisition conditions that maximize the ability to acquire data at the desired resolution with reasonable signal-to-noise and cut stability.

## Embedding and imaging tissue culture monolayers using SBF-SEM

Life science research relies heavily on mammalian cell lines cultured as monolayers *in vitro*. However, it can be challenging to image cell monolayers by SBF-SEM, in large part due to their limited depth which creates a thick layer of non-conductive resin between the cells and the sample pin, and the presence of large areas of non-conductive resin between cells. This is typically not a challenge for the majority of tissue samples due to the density of the tissue (except tissues such as lung, which contain large air spaces without cell structure) and the depth of the tissue. The absence of many conductive paths in these cultured cell samples can result in the buildup of a net negative charge at the sample surface at higher electron doses. The accumulation of negative charge can break the chemical bonds in the resin, which weakens the sample block face [60, 65, 66]. Such weakened resin hinders the sample cutting performance, visible as cut and skip, where the softened resin block is compressed by the knife instead of being cut or as uneven cutting across the block face (chattering). These defects can be mitigated in part by modulating imaging parameters as well as including conductive embedding resins [67]. Conductive resins mitigate the surface charging of SBF-SEM samples by way of allowing a path to ground for accumulated electrons in an otherwise non-conductive material [67]. While this may provide a solution to imaging and cutting artifacts, it is not ideal for many SBF-SEM applications. Any project that pursues *en-face* embedding of monolayers may have difficulty identifying the regions of interest if conductive resins are used as the

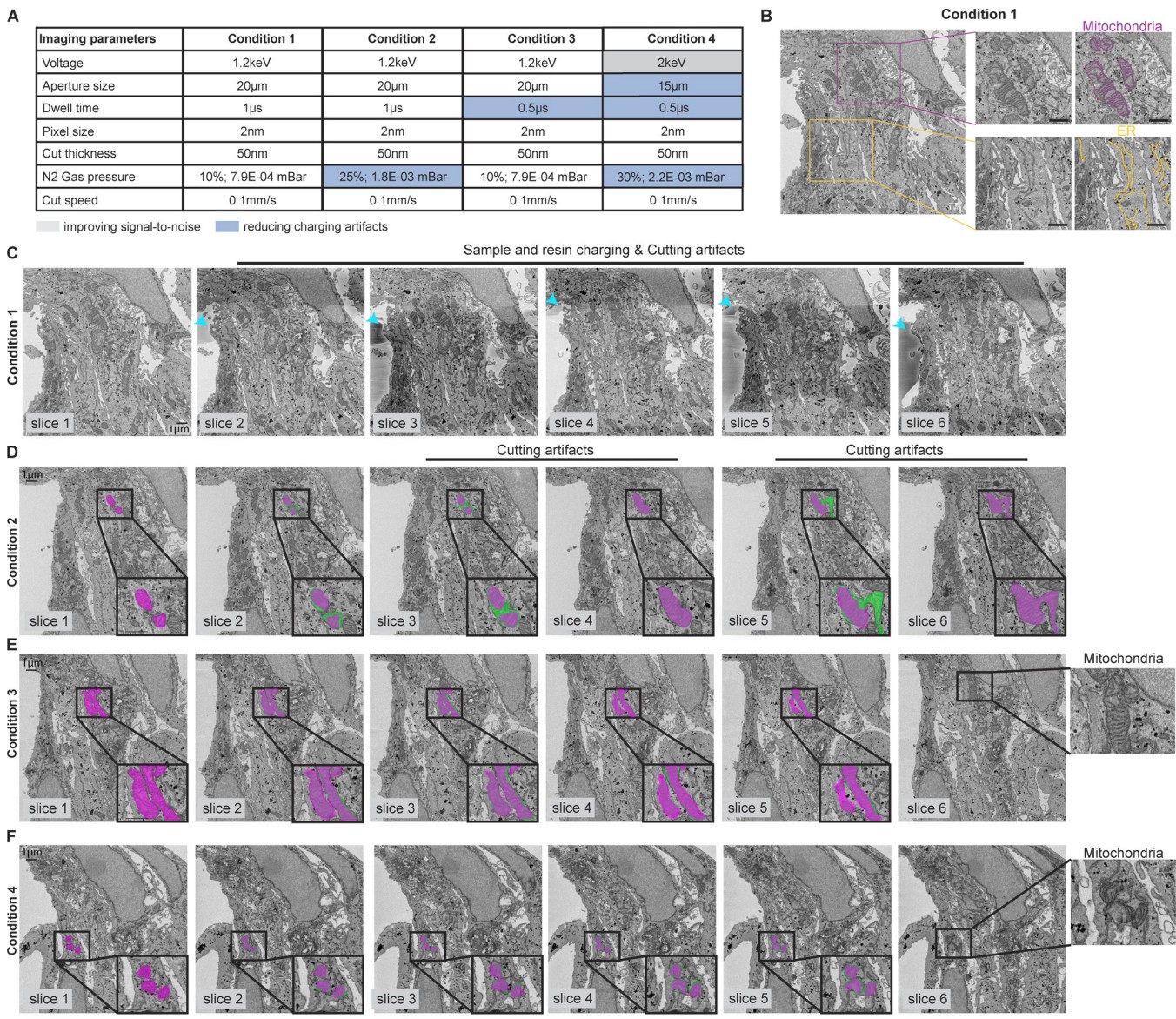

**Fig 4. Effect of different imaging parameters on image quality.** (A) Table showing the imaging parameters tested. (B) A representative slice from the block face. Insets highlight the ultrastructure of different host cell organelles - mitochondria, magenta; ER, yellow. (C) Slices 1–6 from Condition 1. Cyan arrowheads highlight charging and cutting artifacts. (D) Slices 1–6 from Condition 2. Insets show the overlay of mitochondria segmented in magenta (preceding slice) and green. No change in mitochondria morphology is observed from slice 3 to slice 4 and slice 5 to slice 6, highlighted by a complete overlap of the magenta and green annotations. This indicates a cutting defect between slices 3 and 4, and between slices 5 and 6. (E-F) Slices 1–6 from Condition 3 (E) and Condition 4 (F). Magenta and green annotations do not overlap from slice 1 to slice 6 in both these conditions. Continuous changes are expected from one slice to the next, showing that cutting defects do not occur. Also see Movies 1–4. Scale bars: 1 μm.

additives often reduce the transparency of the embedding resin. Therefore, SBF-SEM imaging of cell monolayers requires modifications of standard sample preparation protocols not only in sample staining and embedding, but also on sample mounting methods, as well as an adjustment of the different imaging parameters to ensure robust imaging and data quality. Previously, SBF-SEM has been used to examine the organization of organelles as well as to characterize the life cycle of pathogens in tissue culture monolayers [23, 68–73]. Depending on the biological question being addressed, two sample preparation techniques are particularly

useful: 1) embedding as cell pellets; or 2) *en face* embedding of cell monolayers. In the embedded cell pellet method, adherent cells are first fixed and detached from the cell culture dish, and pelleted, prior to staining and embedding (Fig 2A). In the *en face* embedding method, cells are fixed, stained and embedded directly in the cell culture dish thereby preserving the orientation of the cell (Fig 2B). The cell pellet method is ideal when the underlying orientation of the cell is not important and the biological event being studied is observed frequently (ideally in >50% of cells). Because cells in the pellet are tightly packed together, many cells can be imaged simultaneously. The *en face* method is particularly useful to study rare biological events by correlative microscopy, as it allows users to combine the information from two different microscopy modalities - light microscopy (LM) to identify the rare event, and electron microscopy (SBF-SEM) to image it at a higher resolution for the relevant area of interest. However, with a limited field of view, the imaging area is restricted to just one or a few cells at a time, and therefore the throughput of *en face* embedding and imaging is low. For any given sample, the choice of cell pellet embedding versus *en face* embedding will depend on the underlying biology being examined. A list of advantages and disadvantages for each sample preparation method are described in Table 1.

## Advantages and limitations of SBF-SEM

The advantage of SBF-SEM lies in the ability of this technique to acquire quantitative ultrastructural information on biological samples in an automated manner over large imaging areas and volumes [10, 74, 75]. Image acquisition is fully automated, and serially acquired images are well-aligned, with small amounts of image translation in the x and y that can be corrected post acquisition. Moreover, correlative light and SBF-SEM (CLEM) allows tracking of specific biological events or subcellular location by either widefield or fluorescence microscopy and correlates it to the underlying ultrastructure in the area of interest. Limitations to this technique are similar to all EM ultrastructural analysis, 1) sample preparation takes several days and should be tailored to the biological sample being studied, 2) the use of chemical fixatives and heavy metal staining during sample preparation may alter the properties of the underlying ultrastructure. Alternatively, high pressure freezing and freeze substitution can be considered in order to capture fast cellular events and to have better structural preservation, and 3) z-resolution is limited by the ultramicrotome knife (~25 nm) [71]. Depending on the biological question, other vEM techniques might also be useful to consider, such as FIB-SEM [5] which has greater Z-resolution, array tomography [76], which can be combined with

**Table 1. Table of advantages and disadvantages of the cell pellet versus *en face* embedding of adherent tissue culture cells.**

| Method | Pro | Con |
|---|---|---|
| Cell pellet | Higher throughput (multiple cells or events per sample block) | Complete cells are difficult to capture |
| | Amenable to adherent and suspension cell cultures | Requires that the biological event being studied occurs frequently (>50% of cells) |
| | Increased sample depth for imaging | |
| | Better sample conductivity owing to crowded or densely packed cells pellets | |
| *en face* with CLEM | Identify and capture rare biological events | Lower throughput (1 or a few cells per sample block) |
| | Complete cell can be captured | Poorer conductivity owing to limited sample depth |

protein localization, and TEM-based vEM such as serial section TEM (ssTEM) [77] and serial section electron tomography (ssET) [78] with higher *x-y* resolution.

## Materials and methods

The protocol described in this peer-reviewed article is published on protocols.io (doi.org/10.17504/protocols.io.e6nvwdz5zlmk/v2), and is included for printing as S1 File with this article.

## Results

### Representative results from imaging tissue culture cells by SBF-SEM

Using the protocol described above, a new user should be able to successfully fix, stain, embed, trim, mount and image tissue culture cells using either a cell pellet or an *en face* protocol. In this section we show representative results from imaging a cell pellet embedded sample and an *en face* embedded sample (Fig 5). We have used these protocols to investigate the biology of an

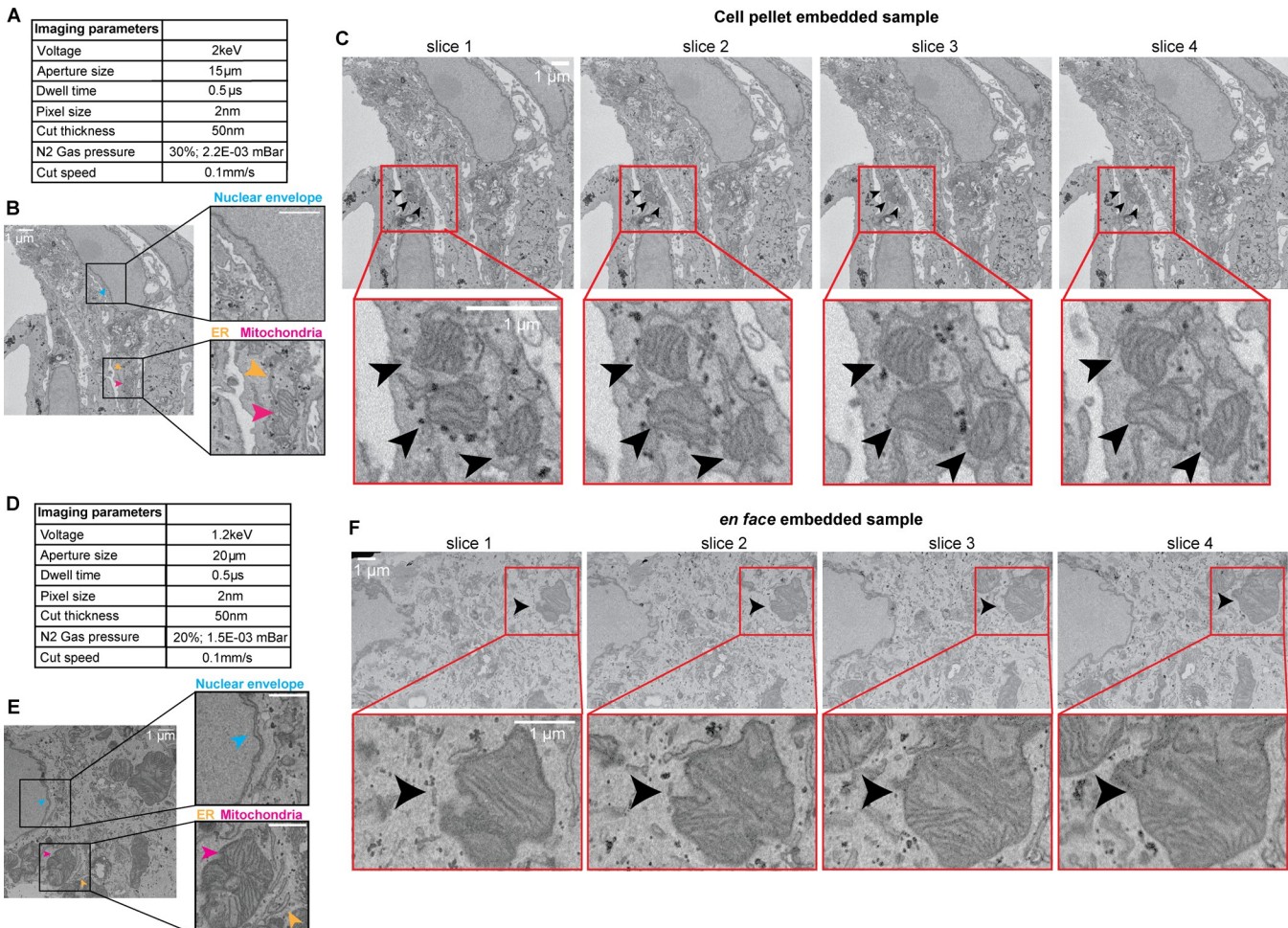

**Fig 5. Representative imaging results for cell pellet and *en face* embedded samples.** (A, D) Tables showing the imaging parameters used to achieve good signal-to-noise and avoid charging artifacts in the cell pellet sample (A) or *en face* sample (D). (B, E) Representative slices from the cell pellet (B) or en face (E) samples using the imaging parameters listed in (A) and (D), respectively. Insets highlight the ultrastructure of different host cell organelles - nucleus, blue arrowhead; mitochondria, magenta arrowhead; ER, yellow arrowhead. (C, F) Slices 1–4 of the cell pellet (C) or en face (F) samples. Also see Movies 5–6. Scale bars 1 μm.

obligate intracellular parasite, *E. intestinalis* in Vero cells [79]. In both the cell pellet and *en face* examples described here, the goal was to acquire images in which the ultrastructures of organelles like the mitochondria and/ or endoplasmic reticulum can be resolved. We used a pixel size of 2 nm. However, a 4–8 nm pixel size would be a reasonable starting point in order to answer a majority of biological questions that require a well-defined organelle structure. The resulting image stacks can be examined using freely available image analysis software, such as Fiji [80], and more complex analyses such as segmentation of cellular structures and 3D reconstruction can be carried out using dedicated software packages, such as Dragonfly [81].

## Supporting information

**S1 File.**
(PDF)

## Acknowledgments

We thank Jason (Xiangxi) Liang from the NYU Microscopy Lab with assistance in the preparation of EM samples. We thank Ari Davydov for critical reading and feedback on the manuscript and all members of the Bhabha / Ekiert labs for helpful discussions. Some portions of the Fig were created with Biorender.com

## Author Contributions

**Conceptualization:** Noelle V. Antao, Damian C. Ekiert, Gira Bhabha.

**Data curation:** Noelle V. Antao, Joseph Sall.

**Formal analysis:** Noelle V. Antao.

**Funding acquisition:** Damian C. Ekiert, Gira Bhabha, Feng-Xia Liang.

**Methodology:** Joseph Sall, Christopher Petzold.

**Resources:** Damian C. Ekiert.

**Supervision:** Damian C. Ekiert, Gira Bhabha, Feng-Xia Liang.

**Visualization:** Noelle V. Antao.

**Writing – original draft:** Noelle V. Antao, Joseph Sall.

**Writing – review & editing:** Noelle V. Antao, Damian C. Ekiert, Gira Bhabha, Feng-Xia Liang.

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
