## [Decision Letter · Decision Letter 0]

20 Dec 2023

PONE-D-23-34295Sample preparation and data collection for Serial Block Face Scanning Electron Microscopy of Mammalian Cell MonolayersPLOS ONE

Dear Dr. Bhabha,

Thank you for submitting your manuscript to PLOS ONE. After careful consideration, we feel that it has merit but does not fully meet PLOS ONE’s publication criteria as it currently stands. Therefore, we invite you to submit a revised version of the manuscript that addresses the points raised during the review process.

We look forward to receiving your revised manuscript.

Kind regards,

Furqan A. Shah

Academic Editor

PLOS ONE

Journal Requirements:

"We gratefully acknowledge the following funding sources: SSP-2018-2737 (Searle Scholars Program, to G.B.), R01AI147131 (NIAID, to G.B.), Irma T. Hirschl Career Scientist Award (to G.B.). G.B. is a Pew Scholar in the Biomedical Sciences, supported by The Pew Charitable Trusts (PEW-00033055). The NYU Microscopy Core is partially supported by NYU Cancer Center Support Grant NIH/NCI P30CA016087, and Zeiss Gemini 300 SEM with 3View was purchased with support of NIH S10 ODO019974."

5. Please expand the acronym “NIAID” (as indicated in your financial disclosure) so that it states the name of your funders in full.

"We thank Jason (Xiangxi) Liang from the NYU Microscopy Core with assistance in the 

preparation of EM samples. We thank Ari Davydov for critical reading and feedback on 

the manuscript and all members of the Bhabha / Ekiert labs for helpful discussions. We 

gratefully acknowledge the following funding sources: SSP-2018-2737 (Searle Scholars 

Program, to G.B.), R01AI147131 (NIAID, to G.B.), Irma T. Hirschl Career Scientist Award 

(to G.B.). G.B. is a Pew Scholar in the Biomedical Sciences, supported by The Pew 

Charitable Trusts (PEW-00033055). The NYU Microscopy Core is partially supported by 

NYU Cancer Center Support Grant NIH/NCI P30CA016087, and Zeiss Gemini 300 SEM 

with 3View was purchased with support of NIH S10 ODO019974"

"We gratefully acknowledge the following funding sources: SSP-2018-2737 (Searle Scholars Program, to G.B.), R01AI147131 (NIAID, to G.B.), Irma T. Hirschl Career Scientist Award (to G.B.). G.B. is a Pew Scholar in the Biomedical Sciences, supported by The Pew Charitable Trusts (PEW-00033055). The NYU Microscopy Core is partially supported by NYU Cancer Center Support Grant NIH/NCI P30CA016087, and Zeiss Gemini 300 SEM with 3View was purchased with support of NIH S10 ODO019974."

Reviewers' comments:

Reviewer's Responses to Questions

**Comments to the Author**

1. Does the manuscript report a protocol which is of utility to the research community and adds value to the published literature?

Reviewer #1: No

Reviewer #2: Yes

Reviewer #3: Yes

Reviewer #4: No

2. Has the protocol been described in sufficient detail?

To answer this question, please click the link to protocols.io in the Materials and Methods section of the manuscript (if a link has been provided) or consult the step-by-step protocol in the Supporting Information files.

The step-by-step protocol should contain sufficient detail for another researcher to be able to reproduce all experiments and analyses.

Reviewer #1: Partly

Reviewer #2: Partly

Reviewer #3: Yes

Reviewer #4: Partly

3. Does the protocol describe a validated method?

Reviewer #1: Yes

Reviewer #2: Yes

Reviewer #3: Yes

Reviewer #4: Yes

4. If the manuscript contains new data, have the authors made this data fully available?

Reviewer #1: N/A

Reviewer #2: N/A

Reviewer #3: Yes

Reviewer #4: N/A

**5. Is the article presented in an intelligible fashion and written in standard English?**

Reviewer #1: Yes

Reviewer #2: Yes

Reviewer #3: Yes

Reviewer #4: Yes

6. Review Comments to the Author

Reviewer #1: The manuscript by Antao and co-authors entitled “Sample preparation and data collection for Serial Block Face Scanning Electron Microscopy of Mammalian Cell Monolayers” is a lab protocol. It intends to orient a novice user in the volumetric microscopy methods and to help such a user to make an informed decision on the exact protocol that best addresses their research objectives. Although the intent is commendable, the manuscript content is banal. The principles of electron-matter interactions are not explained, the volume-resolution dilemma is not even mentioned, the issue of non-isometric voxel is not mentioned as well, and no quantitative metrics are presented in the results, despite the availability of SBF-SEM data to the authors. A novice reader can obtain this level of information by reading Wikipedia or by talking to a technician for 15 minutes.

Detailed comments:

Intro, end of 1st paragraph. This is the place where a passage on the importance of context would fit. The beauty of SBF-SEM microscopy is that a relatively large area can be imaged in every 2D frame, so that the spatial relationships amongst the cells, and between the cells and the extracellular matrix can be analysed in full, guiding the operator towards a higher-resolution method in a correlative manner. Unfortunately, by combining the X-Y voxel of 2 nm with a 50 nm slice thickness the authors generate an inherently astigmatic dataset – in any reconstructed plane (any plane which is not the plane of acquisition) the features would be unresolvable and smeared. And in the plane of acquisition the context is truncated to a tiny square. Such an approach robs the operator of both the context (in the acquisition plane) and of the resolution (in all reconstructed planes).

Overview of the goals: “to provide the best image quality and highest resolution possible”. The first part is trivial – every experimental method strives to provide the best quality, as long as natural limitations permit. The second part is inaccurate – often the largest field of view at a reasonable resolution is the goal. The overall objective of all microscopy methods is rather the harmony between the resolution and the field of view.

Sample preparation section. Specificity of stains – is it your common knowledge, or a more precise account can be provided with a reference to a source (for example, A. Hayat “Electron Microscopy Techniques”)?

TCH and similar binding compounds (glutaraldehyde, tannins) are called mordants.

Where is the osmolarity and pH control mentioned in the preparation sequence?

Cryogenic prep pathway, HPF and FS – this is not only for rare events. Cryogenic preparation serves primarily for delicate preservation of hydrated specimens as close as possible to their native state.

Image acquisition section. What about line- and frame averaging for charging reduction? What about the subsurface interaction volume of the electron beam? Even when the optics are perfect, a beam whose energy is too high will yield BSE/SE from the interaction volume incomparably larger than the desired pixel size, and this is especially germane for biological samples (attributable to their light element composition). Where is your discussion of E2 – the energy at which the incoming and outcoming electrons balance out, and no charging is observed (see Joy and Joy 1996, Low voltage electron microscopy)?

Embedding section. It is unclear how charging results in cutting defects. No conductive resins are mentioned.

Cells fixed in a monolayer – what about compound substrates, such as coated coverslips, or sapphire disks placed in the culture well, for further HPF?

Results. The 2x2x50 voxel makes no sense, but any views different from the plane of acquisition are omitted in the figure. This reviewer observes no difference between the pellet and monolayer cultures. Indeed, to quantify the difference, the volumetric images must be processed and segmented with morphologic metrics provided. How many cells fit in the field of view, what is their morphology, how do their contacts look, what is the distribution of the organelles in the monolayer and pellet specimens? What is the abundance of debris and damaged cells? Is there any evidence of the extracellular matrix elements? Without quantitative analysis this paper can provide very little insight.

Minor comment: numbering pages and lines would be helpful.

Reviewer #2: The manuscript submitted by Noelle V. Antao and colleagues seeks to outline sample preparation and image acquisition for serial block-face scanning electron microscopy imaging of mammalian cell monolayers. The manuscript’s stated purpose is to “highlight the challenges and key steps of sample preparation, and the consideration of imaging variables that will facilitate the acquisition of high quality datasets.” However, in this reviewer’s opinion, the level of detail provided and the depth by which this process is discussed in the primary text is severely lacking. Given the lack of detail and discussion in this manuscript, it is difficult to see its role beyond the methods section provided in the primary author’s most recent publication (DOI: 10.1038/s41467-023-43215-0). The manuscript would benefit from the inclusion of a more in depth discussion of the protocol troubleshooting and optimization methods, expansion of details provided in the main text, and integration of supplemental file “S1_file” with the main text, figures and tables.

Concerns:

1. Introduction mentions the improved resolution of vEM over light microscopy, but spends no time discussing this difference and does not state the resolution limit of light microscopy. In this discussion, it would be necessary to also touch on the resolution limits of super-resolution light microscopic methods that are currently in use.

2. The “target audience” paragraph appears out of place in a methods paper.

3. The authors mention that “the SBF-SEM instrument is comprised of an SEM, an ultramicrotome mounted within the SEM chamber, and the necessary hardware and software to control image acquisition.” This statement is vague and the necessary hardware and software should be covered more thoroughly. If specific hardware or software cannot be stated, the purpose of the hardware/software should be covered.

4. Overview of SBF-SEM: In this section the authors discuss the factors that determine image quality. This appears to be a simplified list, as it does not discuss proper fixation or staining of samples, nor does it mention beam spot size (only pixel size), which is an important factor for image resolution, or the electron beam settings which can determine the focal depth and depth of field.

5. Sample preparation: The authors mention that “a mixture of aldehydes” can be used, but does not provide details on what those mixtures might be or why an individual might choose one over another.

6. Sample preparation: The authors mention that samples can be “trimmed to a shape (truncated pyramid, cube or cuboid) and size that will be amenable to cutting by the ultramicrotome mounted in the SEM chamber” but does not discuss the positives and negatives of various shapes or what a “amenable to cutting” might mean to newcomers to SBF-SEM.

7. Sample preparation: The authors mention that “sample preparation requires[s] optimization at the level of: 1) the combination and concentration of heavy metals, 2) the incubation time of each stain to ensure even stain penetration throughout the sample, 3) the type of embedding resin and 4) the dimensions of the final sample block that is loaded into the SEM chamber.” Nowhere in the manuscript's primary text are methods of optimization discussed, nor are examples given for what this optimization process might entail and how users will know whether their methods are optimized or not. It would strengthen the manuscript considerably if the optimization process was discussed and example images were provided, as is seen in similar publications (e.g., DOI: 10.3791/62045).

8. Image acquisition: “Charging artifacts can be resolved by altering parameters like beam current, dwell time and pixel size during image acquisition.” In what ways should these parameters be altered? What do positive and negative results look like? This section is light on details.

9. Embedding and imaging tissue culture monolayers using SBF-SEM: The authors state that a list of advantages and disadvantages for pellet embedding versus en face embedding are described in the supplementary materials. However, as this is a manuscript focused on vEM of cell monolayers, this information should not be relegated to the supplement and should be expanded upon in the main manuscript body.

10. The section titled “Level of expertise needed to implement this protocol” appears out of place.

11. The materials and methods discussed in the main body of the manuscript are highly specific to a single cell line and experimental conditions conducted in the authors’ laboratory. Given the title of this manuscript, this section should be expanded upon to be much more robust and cater to a much larger range of mammalian cell monolayer systems. I do not see the purpose of this section beyond the methods section provided in the primary author’s most recent publication (DOI: 10.1038/s41467-023-43215-0). The majority of this information is standard in the field and has been covered in a large range of published articles (doi: 10.3390/microorganisms9061194, doi: 10.1242/jcs.188433, doi: 10.3390/v7122940, doi: 10.1016/bs.aivir.2019.07.005, doi: 10.1111/jmi.12667, https://doi.org/10.1016/bs.mcb.2023.01.019).

12. What is the reasoning behind the supplemental file “S1_file” being compiled in the way that it is and why is it relegated to the supplement? It is this reviewer’s opinion that the majority of the information and figures in this file, outside of the step-by-step protocol, should be integrated into the main body text, as these are the details that a large portion of readers will be interested in.

13. The manuscript does not have the required link to the protocol on “protocols.io” in the Materials and Methods section. In addition, Supporting Information file 1 does not contain the required caption, “S1: Step-by-step protocol, also available on protocols.io.”

14. The manuscript mentions a number of embedding resins, but does not provide context in the main text to assist the reader in choosing one.

15. As this process can involve a great deal of optimization and troubleshooting, this manuscript would benefit greatly from a robust discussion on optimization and troubleshooting.

16. Supplemental movies – If possible, please include datasets with a greater number of sections.

Reviewer #3: Comments to the author:

The lab protocol article produced by Anteo et al. explores serial block face preparation techniques for the study of in vitro tissue culture cell samples, notably using serial block face SEM (SBF-SEM). The authors share in a comprehensive manner how this technique can be applied, data optimizing methods, and challenges, to facilitate high-quality data acquisition using SBF-SEM which has broad applications in biological communities and beyond. The submitted work aligns well with the aims and scope of this journal, particularly considering its focus on characterization that has implications for interdisciplinary fields, and I certify that the work seems to meet the journal’s criteria for utility, validation, and availability for lab protocol papers. Therefore, I am of the opinion that this work would be eligible to be published in PLoS ONE and I have minor revisions to suggest for strengthening the published work and how it is framed before publication. This reviewer urges the authors to please address the following points for consideration of revision:

1. I thought the introduction was well framed and wanted to suggest when comparing FIB-SEM and SBF-SEM, the authors should provide a numerical comparison range to describe what “larger volumes” are for SBF-SEM to discuss the differences between the techniques. This could involve discussing differences in volume z depth as well as spatially in X-Y for the field of view as applicable. PFIB-SEM could also be mentioned for the completeness of the introductory discussion.

2. It is suggested that the authors frame the introduction more broadly to the audience about these methods and mention how your team is presenting a very specific protocol of how SPF-SEM can be conducted with specific instruments and chemical fixation and embedding methods, but that other instruments and preparation methods may be used.

3. In the introduction for this sentence “More recently this methodology has been adapted to study the ultrastructure of other biological tissue samples e.g. bone tissue, developing unicellular organisms and cell monolayers.” The authors should list references for each of the listed examples for readers to refer to.

4. I note PLoS One describes acceptable lab protocols for publication as follows: “Lab Protocols describing routine methods or extensions and modifications of routine methods that add little value to the published literature will not be considered for publication.” The article is advanced enough, but the introduction frames it as quite routine. I urge the authors to better highlight within their introduction the value of sharing this comprehensive lab protocol article and that limited detailed protocols are widely available within this field to better showcase the impact/importance of the submitted work.

5. In sample preparation, should electron beam sensitivity and softness be better discussed/addressed briefly?

6. In this sentence: “The goal for vEM imaging techniques such as SBF-SEM is to obtain quantitative volumetric information of biological samples with the best image quality and highest resolution possible.” Is the highest resolution possible the goal? Or would it be more appropriate to say something along the lines of being able to achieve the necessary resolution to visualize key biological phenomena to answer your life science questions?

7. In discussing heavy stains used, I suggest also mentioning to readers Uranyless/UAR-EMS Uranyl Acetate Replacement Stains as an emerging safer alternative to uranyl acetate as an additional option to share.

8. In the section overviewing SBF-SEM, for point #2) I recommend rephrasing the section discussing image resolution, which is typically defined by the smallest resolvable feature and related to probe settings and not necessarily pixel size; moreover, many SEMs can push up to 0.5 nm resolution. Reference 17 here seemed ill-fitted here…

9. I’m not sure if this is a stylistic thing pertinent to the style of this article, but I found that the authors use a lot of lists in paragraphs in their writing which made things sometimes hard to follow. E.g.. the paragraph bridging pages 11-12 (start of sample preparation subsection) includes 3 lists within one paragraph!

10. For image acquisition: does pixel size affect imaging in the way you are discussing related to charging artifacts? Or do you mean to discuss that pushing to certain spatial resolutions at higher magnifications can influence imaging due to beam interactions? I wanted to emphasize that pixel size does not equate necessarily to affecting charging, while often they are related; at the end of the day, pixel size has to do with your camera and magnification and can be influenced simply by checking off binning parameters. Something to consider.

11. In the PDFs and supplemental, the text is hard to read in certain figures and quite small. Particularly for figures 1 and 2 in the supplemental. It is suggested that the authors make the text in figures larger accordingly.

12. One last comment, I felt that post-processing was not discussed in the protocol in great detail. I think readers would benefit from an expansion of how Fiji was used for instance in this work, even if just by a couple sentences and leading readers elsewhere to learn more.

Reviewer #4: This paper begins with a simple introductory opening for the non-expert in SBF and FIB-SEM but then the paper never dives deeper into the reasons for selecting certain parts of the protocol ie the resin or stain. It essentially presents standard methods for cell embedding with little reference to prior work or placing the protocol in context. Therefore, this paper could benefit from substantial expansion to the references and discussion. There’s nothing referenced on the types of resins or stains used, or why. These are critical in the workflow for preparing a sample.

Please reference this sentence, readers will like to have access to these works too : More recently this methodology has been adapted to study the ultrastructure of other biological tissue

samples e.g. bone tissue, developing unicellular organisms and cell monolayers.

Similarly – do you have references to these stains used in SBF or FIB-SEM, it would be helpful for readers to access protocols: These heavy metal salts include: osmium

tetroxide, which preferentially interacts with unsaturated lipids; thiocarbohydrazide, which

binds to osmium in the tissue and acts as a bridging agent to which more osmium can

bind, thereby enhancing the contrast of lipid components in the cell; uranyl acetate, which

stains nucleic acids and proteins an provides general contrast to biological samples; lead

aspartate, which interacts with membranes and proteins.

Some discussion on the type of resin used for embedding (Durcupan, Embed 812, Epon, PMMA, etc) would be particularly useful – there are several with varying hardness, could you refer readers to more information on this (given it wasn’t the subject of this investigation)? Are any particularly well suited to cells?

Fig 2: FCC – define in caption

No references are provided in the “Image Acquisition” section – there are papers to highlight the effect of these settings on imaging – it would be worthwhile to reference some. These seem like novice tips on basic SEM operation.

CLEM is mentioned in the intro and abstract, but nothing is really presented on this in the paper. Either include more or remove reference to it being demonstrated.

What was different between the samples prepared in different ways? Since different imaging parameters were used, it's also hard to make any conclusions – can you comment on any pros/cons of either method – as you observed in what's shown in Fig 3 (not just in general)?

7. PLOS authors have the option to publish the peer review history of their article (what does this mean?). If published, this will include your full peer review and any attached files.

Reviewer #1: No

Reviewer #2: No

Reviewer #3: No

Reviewer #4: No

---

## [Author Response · Author response to Decision Letter 0]

2 Feb 2024

We are grateful for the comments from all four reviewers, which have helped us to improve our manuscript. Referee comments are in black and our responses are in blue. All changes in the revised manuscript are also in blue.

We would like to start by clarifying that our manuscript is intended for the Lab Protocols format at PlosOne https://collections.plos.org/collection/lab-protocols/ [“PLOS ONE is committed to pushing the boundaries of Open Science by improving the reproducibility and transparency of scientific research. Earlier this year we launched Lab Protocols in collaboration with protocols.io. Lab Protocols offer authors the opportunity to share their peer reviewed step-by-step protocols with the community whilst receiving credit for their contributions. In this collection, we are highlighting the first set of Lab Protocols that PLOS ONE has published from across our broad scope, from molecular biology, biotechnology, structural biology to archaeology.”] We have adhered to the format of this type of article, in which the step-by-step protocol is the main feature. The protocol is available on protocols.io (dx.doi.org/10.17504/protocols.io.e6nvwdz5zlmk/v2) and is also referenced extensively in the manuscript. In our understanding, S1_File is a duplicate of the step-by-step protocol, which we have also provided. Our goal is to share a comprehensive protocol for SBF-SEM of mammalian cells accessible to novice users, which is surprisingly hard to find in the field, as noted by Reviewer 3. In the manuscript, our goal is to familiarize users with the technique and methodology sufficiently to follow the protocol and understand critical steps. With this in mind, we have made revisions to the manuscript, and provide a point-by-point response below. 

Reviewer #1: The manuscript by Antao and co-authors entitled “Sample preparation and data collection for Serial Block Face Scanning Electron Microscopy of Mammalian Cell Monolayers” is a lab protocol. It intends to orient a novice user in the volumetric microscopy methods and to help such a user to make an informed decision on the exact protocol that best addresses their research objectives. Although the intent is commendable, the manuscript content is banal. The principles of electron-matter interactions are not explained, the volume-resolution dilemma is not even mentioned, the issue of non-isometric voxel is not mentioned as well, and no quantitative metrics are presented in the results, despite the availability of SBF-SEM data to the authors. A novice reader can obtain this level of information by reading Wikipedia or by talking to a technician for 15 minutes.

Detailed comments:

Intro, end of 1st paragraph. This is the place where a passage on the importance of context would fit. The beauty of SBF-SEM microscopy is that a relatively large area can be imaged in every 2D frame, so that the spatial relationships amongst the cells, and between the cells and the extracellular matrix can be analysed in full, guiding the operator towards a higher-resolution method in a correlative manner. Unfortunately, by combining the X-Y voxel of 2 nm with a 50 nm slice thickness the authors generate an inherently astigmatic dataset – in any reconstructed plane (any plane which is not the plane of acquisition) the features would be unresolvable and smeared. And in the plane of acquisition the context is truncated to a tiny square. Such an approach robs the operator of both the context (in the acquisition plane) and of the resolution (in all reconstructed planes).

We thank the reviewer for their comment and have amended the text at the end of the first paragraph. It now reads ‘While the z resolution is higher using FIB-SEM, the milling process is slower, which can, in practice, limit the total area and volume that can be imaged in a given time relative to SBF-SEM. The ultramicrotome/diamond knife used for sectioning in SBF-SEM makes the cutting of sections about 4 times faster, allowing for the acquisition of total volumes of 106 μm3 [9,10]. In contrast, volumes analyzed by ion beam milling are typically 103 μm3 [11,12]. The faster cutting enables efficient imaging of samples with larger volumes by SBF-SEM, and allows users to evaluate the relationship between cells over relatively large fields of view, across all imaged planes [4,13–15]. It should be noted that the use of plasma FIB (pFIB) with other ion species like argon [16] and xenon [17] may provide finer and faster milling when compared to gallium, thereby increasing the milling speed and surface area analyzed by pFIB-SEM.’ (Page 3).

Overview of the goals: “to provide the best image quality and highest resolution possible”. The first part is trivial – every experimental method strives to provide the best quality, as long as natural limitations permit. The second part is inaccurate – often the largest field of view at a reasonable resolution is the goal. The overall objective of all microscopy methods is rather the harmony between the resolution and the field of view.

We agree with the reviewer and have adjusted the language in the text to reflect that. The sentence now reads ‘The goal for vEM imaging techniques such as SBF-SEM is to obtain volumetric information of biological samples with the best image quality and at a resolution sufficient to answer the biological question of interest.’ (Page 4)

Sample preparation section. Specificity of stains – is it your common knowledge, or a more precise account can be provided with a reference to a source (for example, A. Hayat “Electron Microscopy Techniques”)?

We have now included a more comprehensive list of references for various mordants and heavy metal stains. This is a list of references added:

Deerinck TJ, Bushong EA, Thor AK, Ellisman M, Deerinck TJ, Bushong EA, et al. NCMIR methods for 3D EM: a new protocol for preparation of biological specimens for serial block face scanning electron microscopy. 2010.

Seligman AM, Wasserkrug HL, Hanker JS. A new staining method (OTO) for enhancing contrast of lipid--containing membranes and droplets in osmium tetroxide--fixed tissue with osmiophilic thiocarbohydrazide(TCH). J Cell Biol. 1966;30: 424–432.

Willingham MC, Rutherford AV. The use of osmium-thiocarbohydrazide-osmium (OTO) and ferrocyanide-reduced osmium methods to enhance membrane contrast and preservation in cultured cells. J Histochem Cytochem. 1984;32: 455–460.

Wilke SA, Antonios JK, Bushong EA, Badkoobehi A, Malek E, Hwang M, et al. Deconstructing complexity: serial block-face electron microscopic analysis of the hippocampal mossy fiber synapse. J Neurosci. 2013;33: 507–522.

(Eric) Hayat MA. Fixation for Electron Microscopy. Elsevier; 2012.

Sabatini DD, Bensch K, Barrnett RJ. Cytochemistry and electron microscopy. The preservation of cellular ultrastructure and enzymatic activity by aldehyde fixation. J Cell Biol. 1963;17: 19–58.

Jones D, Gresham GA. Reaction of formaldehyde with unsaturated fatty acids during histological fixation. Nature. 1966;210: 1386–1388.

Khan AA, Riemersma JC, Booij HL. The reactions of osmium tetroxide with lipids and other compounds. J Histochem Cytochem. 1961;9: 560–563.

Kelley RO, Dekker RA, Bluemink JG. Ligand-mediated osmium binding: its application in coating biological specimens for scanning electron microscopy. J Ultrastruct Res. 1973;45: 254–258.

Stoeckenius W. Electron microscopy of DNA molecules “stained” with heavy metal salts. J Biophys Biochem Cytol. 1961;11: 297–310.

Terzakis JA. Uranyl acetate, a stain and a fixative. J Ultrastruct Res. 1968;22: 168–184.

Hayat MA. Principles and Techniques of Electron Microscopy: Biological Applications. University Park Press; 1981.

Walton J. Lead aspartate, an en bloc contrast stain particularly useful for ultrastructural enzymology. J Histochem Cytochem. 1979;27: 1337–1342.

Hayat MA. Stains and Cytochemical Methods. Springer Science & Business Media; 1993.

Nakakoshi M, Nishioka H, Katayama E. New versatile staining reagents for biological transmission electron microscopy that substitute for uranyl acetate. J Electron Microsc . 2011;60: 401–407.

Kuipers J, Giepmans BNG. Correction to: Neodymium as an alternative contrast for uranium in electron microscopy. Histochem Cell Biol. 2020;154: 683.

TCH and similar binding compounds (glutaraldehyde, tannins) are called mordants.

We thank the reviewer and have added this into the manuscript (Page 5).

Where is the osmolarity and pH control mentioned in the preparation sequence?

The pH and osmolarity can be found in the materials section of the step-by-step protocol in the S1_File and on protocols.io

Cryogenic prep pathway, HPF and FS – this is not only for rare events. Cryogenic preparation serves primarily for delicate preservation of hydrated specimens as close as possible to their native state.

We apologize for the lack of clarity on this point in the original manuscript. The sentence now reads ‘Special fixation and sample preparation methods such as high pressure freezing and freeze substitution should be considered to analyze fast cellular events, and for better preservation of cell ultrastructure. These methods result in much less sample shrinkage [55–58] and ensure the preservation of samples, as close to their native state as possible.’ (Page 6).

Image acquisition section. What about line- and frame averaging for charging reduction? What about the subsurface interaction volume of the electron beam? Even when the optics are perfect, a beam whose energy is too high will yield BSE/SE from the interaction volume incomparably larger than the desired pixel size, and this is especially germane for biological samples (attributable to their light element composition). Where is your discussion of E2 – the energy at which the incoming and outcoming electrons balance out, and no charging is observed (see Joy and Joy 1996, Low voltage electron microscopy)?

We thank the reviewer for their comment and have added these concepts to the discussion in the image acquisition section, along with relevant refs (Pages 6-7). 

Embedding section. It is unclear how charging results in cutting defects. No conductive resins are mentioned.

The following clarification is now included in the manuscript (Page 8-9).

‘However, it can be challenging to image cell monolayers by SBF-SEM, in large part due to their limited depth which creates a thick layer of non-conductive resin between the cells and the sample pin, and the presence of large areas of non-conductive resin between cells. This is typically not a challenge for the majority of tissue samples due to the density of the tissue (except lung tissue) and the depth of the tissue. The absence of many conductive paths in these cultured cell samples can result in the buildup of a net negative charge at the sample surface at higher electron doses. This accumulation of negative charge can break the chemical bonds in the resin, weakening the sample block face [59,65,66]. Such weakened resin hinders the sample cutting performance, visible as cut and skip, where the softened resin block is compressed by the knife instead of being cut or as uneven cutting across the block face (chattering). These defects can be mitigated in part by modulating imaging parameters as well as including conductive embedding resins [67]. Conductive resins mitigate the surface charging of SBF-SEM samples by way of allowing a path to ground for accumulated electrons in an otherwise non-conductive material [67]. While this may provide a solution to imaging and cutting artifacts, it is not ideal for all SBF-SEM applications. Any project that pursues en-face embedding of monolayers may have difficulty identifying the regions of interest if conductive resins are used as the additives often reduce the transparency of the embedding resin.‘

Cells fixed in a monolayer – what about compound substrates, such as coated coverslips, or sapphire disks placed in the culture well, for further HPF?

It was unclear to us what specifically the reviewer would like us to do in response to this comment. We have added a reference to a review that discusses sample preparation further, but we feel that an extensive discussion on this point is beyond the scope of our protocol manuscript, if we understood the comment correctly. 

Results. The 2x2x50 voxel makes no sense, but any views different from the plane of acquisition are omitted in the figure. This reviewer observes no difference between the pellet and monolayer cultures. Indeed, to quantify the difference, the volumetric images must be processed and segmented with morphologic metrics provided. How many cells fit in the field of view, what is their morphology, how do their contacts look, what is the distribution of the organelles in the monolayer and pellet specimens? What is the abundance of debris and damaged cells? Is there any evidence of the extracellular matrix elements? Without quantitative analysis this paper can provide very little insight.

Several of the answers to these questions depend on the sample and the goal of imaging. For the cell pellet, 10-20 cells can fit a field of view (FOV), if the goal is to obtain a larger area, and if lower resolution is sufficient. For higher resolution imaging, which was our intention in this case, 2-3 cells can be obtained in the FOV. For en face embedded cells, a single cell is usually picked as FOV. However, 2 or 3 ROIs can be imaged simultaneously. We have added this information to the manuscript for our particular case (Page 9).

As noted at the beginning, our goal in this manuscript is to provide a step-by-step protocol, with a rationale and sufficient information on the methodology for the protocol to be useful, which is in line with the Lab Protocols format. As such, we agree with the reviewer that this manuscript does not provide biological insights to the reader. The primary manuscript that used this protocol contains substantial biological insight and analysis (PMID: 37996434).

Minor comment: numbering pages and lines would be helpful

We have numbered the pages. 

Reviewer #2: The manuscript submitted by Noelle V. Antao and colleagues seeks to outline sample preparation and image acquisition for serial block-face scanning electron microscopy imaging of mammalian cell monolayers. The manuscript’s stated purpose is to “highlight the challenges and key steps of sample preparation, and the consideration of imaging variables that will facilitate the acquisition of high quality datasets.” However, in this reviewer’s opinion, the level of detail provided and the depth by which this process is discussed in the primary text is severely lacking. Given the lack of detail and discussion in this manuscript, it is difficult to see its role beyond the methods section provided in the primary author’s most recent publication (DOI: 10.1038/s41467-023-43215-0). The manuscript would benefit from the inclusion of a more in depth discussion of the protocol troubleshooting and optimization methods, expansion of details provided in the main text, and integration of supplemental file “S1_File” with the main text, figures and tables.

Concerns:

1. Introduction mentions the improved resolution of vEM over light microscopy, but spends no time discussing this difference and does not state the resolution limit of light microscopy. In this discussion, it would be necessary to also touch on the resolution limits of super-resolution light microscopic methods that are currently in use.

We thank Reviewer 2 for their suggestion and have included this sentence in the introduction: “Volume electron microscopy (vEM) stands out as a powerful tool capable of producing three dimensional (3D) reconstructions of biological samples, spanning from tissues to individual cells. Resolutions resulting from vEM techniques exceed those attainable with super-resolution microscopy, which can achieve sub-100 nm lateral resolution [1,2]-[3]. Typically, sub-organellar structural features, such as the cristae of mitochondria, can be well resolved using vEM techniques [4].” (Page 3)

2. The “target audience” paragraph appears out of place in a methods paper.

We agree and have removed the target audience paragraph from the introduction.

3. The authors mention that “the SBF-SEM instrument is comprised of an SEM, an ultramicrotome 

---

## [Decision Letter · Decision Letter 1]

14 Mar 2024

Sample preparation and data collection for Serial Block Face Scanning Electron Microscopy of Mammalian Cell Monolayers

PONE-D-23-34295R1

Dear Dr. Bhabha,

We’re pleased to inform you that your manuscript has been judged scientifically suitable for publication and will be formally accepted for publication once it meets all outstanding technical requirements.

Kind regards,

Furqan A. Shah

Academic Editor

PLOS ONE

Additional Editor Comments:

My decision of Accept was based on the assessment that concerns raised by Reviewer #1 were pertaining to theoretical aspects of electron optics and imaging, and were mostly beyond the scope of a Lab Protocol manuscript. Yes, as Reviewer #1 has pointed out that the authors have not shown any 3D visualisation, that is true. However, 2D visualisation of sequential slice-and-view methodology is still a key starting point.

Reviewers' comments:

Reviewer's Responses to Questions

**Comments to the Author**

1. Does the manuscript report a protocol which is of utility to the research community and adds value to the published literature?

Reviewer #1: No

Reviewer #2: Yes

Reviewer #3: Yes

2. Has the protocol been described in sufficient detail?

To answer this question, please click the link to protocols.io in the Materials and Methods section of the manuscript (if a link has been provided) or consult the step-by-step protocol in the Supporting Information files.

The step-by-step protocol should contain sufficient detail for another researcher to be able to reproduce all experiments and analyses.

Reviewer #1: Partly

Reviewer #2: Yes

Reviewer #3: Yes

3. Does the protocol describe a validated method?

Reviewer #1: Yes

Reviewer #2: Yes

Reviewer #3: Yes

4. If the manuscript contains new data, have the authors made this data fully available?

Reviewer #1: N/A

Reviewer #2: Yes

Reviewer #3: Yes

**5. Is the article presented in an intelligible fashion and written in standard English?**

Reviewer #1: Yes

Reviewer #2: Yes

Reviewer #3: Yes

6. Review Comments to the Author

Reviewer #1: The manuscript has marginally improved, but yet contains numerous errors re electron-substrate interaction. For example, the spot size depends on the electron source, electron optics, and electron current (limited by the aperture) and not by the voltage. The unfortunate effect of charging besides the degradation of the substrate, degrades the image quality and masks/distorts the features of interest. Fixative and mordants are different things. Neutral charging is an unusual expression. Overall, the manuscript still contains factual errors (obviously, a bad thing) and no 3D quantitative data on the substrate (a weakness undermining the very purpose of 3D imaging) and as such will do a disservice to the readers. The authors should consult a qualified electron microscopist or physicist to bring the paper to the publishable quality.

Reviewer #2: This manuscript resubmission by Noelle Antao and colleagues, titled "Sample preparation and data collection for Serial Block Face Scanning Electron Microscopy of Mammalian Cell Monolayers" adequately addresses the concerns of this reviewer through extensive revision of the text.

Reviewer #3: I thank the authors for their comprehensive response to my revision points and the revision points of the other reviewers, which were quite extensive! I believe the authors went above and beyond in their revision for a protocol paper considering the guidelines provided by PLOS ONE in my opinion, particularly after surveying other similar-in-style protocols published. I strongly suggest accepting the article for publication now that it is greatly improved.

7. PLOS authors have the option to publish the peer review history of their article (what does this mean?). If published, this will include your full peer review and any attached files.

Reviewer #1: No

Reviewer #2: No

Reviewer #3: No

---

## [Editor Report · Acceptance letter]

31 Jul 2024

PONE-D-23-34295R1 

PLOS ONE

Dear Dr. Bhabha, 

I'm pleased to inform you that your manuscript has been deemed suitable for publication in PLOS ONE. Congratulations! Your manuscript is now being handed over to our production team.

Kind regards, 

on behalf of

Dr. Furqan A. Shah 

Academic Editor

PLOS ONE